

# Accuracy of triggering receptor expressed on myeloid cells 1 in diagnosis and prognosis of acute myocardial infarction: a prospective cohort study

Zhenjun Ji, Rui Zhang, Mingming Yang, Wenjie Zuo, Yuyu Yao, Yangyang Qu, Yamin Su, Zhuyuan Liu, Ziran Gu and Genshan Ma

Department of Cardiology, Zhongda Hospital, School of Medicine, Southeast University , Nanjing, Jiangsu, China

Corresponding author
Genshan Ma, 101010771@seu.edu.cn

## ABSTRACT

**Background:** Acute myocardial infarction (AMI) is one of the fatal cardiac emergencies. The detection of triggering receptor expressed on myeloid cells 1 (TREM1), a cell surface immunoglobulin that amplifies pro-inflammatory responses, screened by bioinformatics was shown to be significant in diagnosing and predicting the prognosis of AMI.

**Methods:** GSE66360, GSE61144 and GSE60993 were downloaded from the Gene Expression Omnibus (GEO) database to explore the differentially expressed genes (DEGs) between AMI and control groups using R software. A total of 147 patients in total were prospectively enrolled from October 2018 to June 2019 and divided into two groups, the normal group ($n = 35$) and the AMI group ($n = 112$). Plasma was collected from each patient at admission and all patients received 6-month follow-up care.

**Results:** According to bioinformatic analysis, TREM1 was an important DEG in patients with AMI. Compared with the normal group, TREM1 expression was markedly increased in the AMI group ($p < 0.001$). TREM1 expression was positively correlated with fasting plasma glucose (FPG), glycosylated hemoglobin (HbAC), and the number of lesion vessels, although it had no correlation with Gensini score. TREM1 expression in the triple-vessels group was significantly higher than that of the single-vessel group ($p < 0.05$). Multiple linear regression showed that UA and HbAC were two factors influencing TREM1 expression. The ROC curve showed that TREM1 had a diagnostic significance in AMI ($p < 0.001$), especially in AMI patients without diabetes. Cox regression showed increased TREM1 expression was closely associated with 6-month major adverse cardiac events (MACEs) ($p < 0.001$).

**Conclusions:** TREM1 is a potentially significant biomarker for the diagnosis of AMI and may be closely associated with the severity of coronary lesions and diabetes. TREM1 may also be helpful in predicting the 6-month MACEs after AMI.

## INTRODUCTION

The mortality rates of cardiovascular diseases continues to rise in Chinese cities and rural areas (*Hu et al., 2019*). As one of the most important cardiovascular emergencies, acute myocardial infarction (AMI) is generally divided into ST-segment elevated myocardial infarction (STEMI) and Non-STEMI (NSTEMI) (*Thygesen et al., 2018*). Early diagnosis and treatment are critical to reducing the incidence of major adverse cardiac events (MACEs) in AMI patients. Currently, clinical diagnosis of AMI still depends on the dynamic change of myocardial infarction (MI) biomarkers and electrocardiogram (ECG). The outcome of patients with AMI is usually predicted by age, left ventricular function, the severity of coronary artery stenosis, and myocardial ischemia (*Loscalzo, 2013*). Hence, it is important to explore new effective biomarkers for the diagnosis and prognosis of AMI.

The rise and development of genomics, proteomics, metabolomics, and transcriptomics have created a large amount of data for clinical and basic research. Bioinformatics methods can be used for secondary analysis and processing of existing bioinformatics resources, such as gene function annotation, pathway analysis, and enrichment analysis, to better carry out life science research. We searched existing blood sequencing datasets of patients with AMI in Pubmed and screened the appropriate differentially expressed genes (DEGs) using bioinformatics methods. Combined with the latest literature progress, Triggering receptor expressed on myeloid cells 1 (TREM1) and toll-like receptor 4 (TLR4) were finally screened as the important DEGs between patients with AMI and healthy controls, and verified them in clinical samples.

The TREM1 gene is located on human chromosome 6 and expressed as two isoforms: membrane-bound TREM1 and soluble TREM1 (sTREM1) (*Jeremie et al., 2015*). TREM1 is an immune receptor expressed in human neutrophils, monocytes, and macrophages; its expression may be related to the activation of Toll-like receptors (TLR) (*Jeremie et al., 2015*). *Boufenzer et al. (2015)* studied the role of TREM1 in AMI based on animal models, which showed that inhibiting TREM1 activation may reduce the inflammatory response after myocardial infarction. Furthermore, they also reported that a new synthetic short peptide, LR12, can inhibit the expression of TREM1 and reduce neutrophil aggregation, monocyte chemokine production, and improve cardiac function after myocardial infarction in mice and rats (*Boufenzer et al., 2015*). *Kutikhin et al. (2016)* found that specific TLR was associated with the polymorphism of the TREM1 gene (including the C/T genotype of TLR4 rs4986791) and the severity of atherosclerosis in the Russian population. TREM1 mediated pathways such as the IL2 pathway, MHC I-mediated APC, and the Toll-like receptors pathway were all involved in the process of innate immunity inflammation after the onset of AMI (*Ormsby et al., 2011*; *Tessarz & Cerwenka, 2008*). However, the significance of TREM1 for guiding clinical diagnosis and treatment of cardiovascular disease still requires further study.

This study aims to explore the role of TREM1 in the diagnosis of MI and prediction of major adverse cardiac events (MACEs).

## MATERIALS AND METHODS

### Screening DEGs

The data expression files GSE66360, GSE61144 and GSE60993 were screened and downloaded from the Gene Expression Omnibus (GEO) database (http://www.ncbi.nlm.nih.gov/geo). The GSE61144 dataset, including seven patients with STEMI and 10 patients with normal control, aimed to identify early significant serum biomarkers of STEMI. GPL6106 Sentrix Human-6 v2 Expression BeadChip was the sequencing platform for GSE61144. The GSE66360 dataset, including 49 patients with STEMI and 50 healthy cohorts, aimed to present the specific gene expression pattern in whole blood of patients and utilized the HG-133U_PLUS_2 microarray sequencing platform. The GSE60993 dataset, including seven STEMI patients and seven healthy patients, aimed to study the molecular signature of acute coronary syndrome (ACS) based on blood transcriptome and to identify novel serum biomarkers for early-stage STEMI. The Illumina HumanWG-6 v3.0 expression BeadChip sequence platform was used for GSE60993. The affyPLM package was applied for data quality control. GSE61144/66360/60993 were treated with limma package in R software to explore the differentially expressed genes (DEGs) between STEMI and control groups. The threshold of differential genes had a false discovery rate (FDR) <0.05 and |logFC| > 1. A Venn diagram drawn using an online tool (http://bioinformatics.psb.ugent.be/webtools/Venn/) generated the common genes intersection of different datasets. Protein-protein interaction (PPI) network construction was performed for functional proteins association analysis using String database (http://string-db.org/), which predicted the physical and functional association among specific proteins. cytoHubba plugin in Cytoscape (https://cytoscape.org/) was used for node network topology analysis of PPI network and pivotal nub genes were screened according to Degree.

### Study design

This study was a prospective, case-control, and cohort study. Patients with AMI or with normal coronary arteries were enrolled and prospectively followed up for 6 months to study the diagnostic and prognostic roles of TREM1.

### Setting

The study was approved by the Ethics Committee of Zhongda Hospital Southeast University (2018ZDSYLL134-P01). All patients gave written informed consents. Patients with AMI or normal coronary arteries were consecutively enrolled from October 2018 to June 2019 in Zhongda Hospital Southeast University according to the inclusion and exclusion criteria. Laboratory and imaging examination data were collected. All enrolled patients received at least 6-months follow-up.

### Participants

In total, 147 patients were enrolled and divided into the normal group ($n = 35$) and the AMI group ($n = 112$). Inclusion criteria: (1) Willing to participate in the study and giving

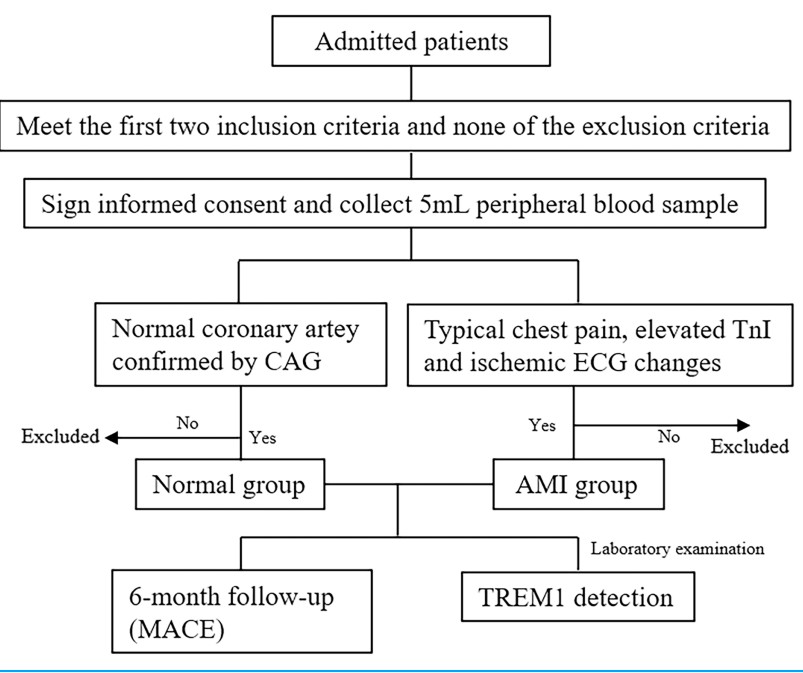

**Figure 1 The flow chart of this study.**

informed consent; (2) Male or non-pregnant female aged 18–85; (3) Normal group: patients with normal coronary arteries confirmed by coronary angiography (CAG). AMI group: typical chest pain, elevated TnI, and ECG showed ischemic ST-T changes. Those who met all the criteria above were selected. Exclusion criteria: (1) Previous coronary artery bypass grafting history; (2) Severe liver or kidney disease (not caused by heart disease); (3) Uncontrolled severe hypertension (systolic blood pressure >180 mmHg and diastolic blood pressure >100 mmHg after standard treatment); (4) Ischemic stroke within one week, previous history of intracranial hemorrhage, gastrointestinal bleeding within 6 months, or major surgery within 30 days; (5) Severe dyspnea, such as bronchial asthma and chronic obstructive pulmonary disease; (6) Suffering from other severe diseases and having a life expectancy less than half a year; (7) Pathological sinus node syndrome, Grade II or III atrioventricular block, or previous syncope history due to bradycardia; (8) Pregnant or lactating women; (9) Other circumstances, as determined by the researchers, which would have made the patient unsuitable for participation in this study. Any patients who did not meet any exclusion criteria were selected. The flow chart was shown in Fig. 1.

## Variables and data sources

Basic laboratory and imaging results included routine blood tests, blood biochemistry tests, fibrinolytic testing, and echocardiography. Basic characteristics included age, gender, history of smoking, diabetes and hypertension, height and weight, and systolic blood pressure. Count of red blood cells (RBC) and white blood cells (WBC), hemoglobin (Hb), alanine aminotransferase (ALT), aspartate aminotransferase (AST), total cholesterol (TC), total triglyceride (TG), albumin (ALB), bilirubin, blood urea nitrogen (BUN), uric acid

(UA), fasting plasma glucose (FPG), glycosylated hemoglobin (HbAC), C-reactive protein (CRP), BNP, and left ventricular ejected fraction (LVEF) were all collected. The enrolled MI patients were divided into single-vessel, double-vessels, and triple-vessels groups according to the number of lesion vessels with >50% stenosis in the left main, left anterior descending, left circumflex, and right coronary arteries. Gensini score calculation was performed following *Gensini's (1983)* published guidelines. Follow-up major adverse cardiovascular events include recurrent arrhythmia, target vessel revascularization (TVR), recurrent myocardial infarction, cardiogenic death, and all-cause death (*Kip et al., 2008*).

## Bias

Bias is a systematic error caused by the fact that the results obtained from the comparative study of each sample group cannot truthfully reflect the real results in the target population. For the systematic error in the selection of study objects-selection bias, we used the medical records system, telephone, and other methods for the follow-up of all patients, to avoid loss of follow-up and reduce bias as far as possible.

We selected the control group from the cardiology department at the same time as the AMI patients in order to reduce differences between the two groups. For the information bias, which mainly occurs in the implementation stage of observation and measurement, we used quantifiable objective indicators as far as possible, while excluding the information bias caused by subjective guesswork. Statistical machines and operating procedures were used in laboratory testing and enzyme-linked immunosorbent assay (ELISA) testing to avoid large information bias. To reduce the confounding biases that can occur in both the design and analysis phases, multivariate analysis and Cox regression were used during data analysis for statistical correction.

## Study size

The clinical sample size was calculated using PASS15.0 software. Since we could not consult the relevant literature on the expression of TREM1 in plasma, we adjusted the sample size through the pre-experiment. The TREM1 expression in the normal group was $127.1 \pm 50.1$ pg/ml ($n = 16$) and in the AMI group was $180.2 \pm 70$ ($n = 60$) (Table S1). According to the control: case = 1:4 (Two-Sample T-Tests) and taking the rate of loss of follow-up < 20% into consideration, it was calculated to include 113 cases. Since TREM1 indicators are rarely reported in the clinical literature, and to further reduce bias, we eventually included 147 samples.

## Sample collection and the enzyme-linked immunosorbent assay (ELISA)

Five mL of peripheral blood was collected from each patient within 4 h of admission. Blood samples were collected by EDTA anticoagulant tubes, stored in refrigerators at 4 °C, and processed within 2 h. The samples were centrifuged at 3000 r/min and 4 °C for 30 min. The plasma was collected and stored in −80 °C refrigerators. ELISA was performed by TREM1 Elisa kit (CUSABIO, China). Anti-TREM1 was coated on the enzyme-label plates. A total of 100 µl plasma was added into each well and incubated at 37 °C incubator

(JINGHONG, Shanghai) for 1 h. Afterwards, the plates were washed with phosphate buffered saline tween-20 (PBST) five times, 30 s each time. Then 100 μl Biotin-antibody was added into each well and incubated at 37 °C for 1 h, and then washed for another 30 s × 3 times. Next, we added 100 μl HRP-avidin to each well which was incubated again for 1 h at 37 °C. A total of 90 μl TMB Substrate was added into each well and incubated for 30 min at 37 °C in a dark environment. Then 50 μl Stop Solution was added into each well, which was read using Microplate Reader (RAYTO RT-6000, Shenzhen, China) at 450 nm within 5 min.

## Statistical methods

Data was processed by SPSS 23.0 and GraphPad Prism 7.0. T-test was used for comparison between two groups and Anova analysis was used for comparison of multiple groups. Pearson correlation analysis and multiple linear regressions were performed to analyze factors affecting variables. Univariate and variate Cox regressions were used for survival and prognosis studies. If the data did not satisfy normality and homogeneity of variance (SD ≥ χ), median and quartile (25%, 75%) were calculated. Variables included in the variate regression were either confirmed clinically relevant or validated by univariate analysis.

# RESULTS

## Candidate genes from screening and PPI network construction

A total of 323, 128 and 147 significantly up-regulated DEGs were respectively screened out in GSE66360, GSE61144 and GSE60993 datasets where 86, 42 and 25 down-regulated DEGs were also found, respectively. The data was graphed into volcano plots (Fig. 2). After the three datasets were integrated, we found 29 common up-regulated and 3 down-regulated genes (Table 1). The final 32 screened genes were submitted to String database to construct an interaction network (Fig. 2), and then cytoHubba was used to analyze hub genes. TLR4 (Degree:11.0), TREM1 (Degree:9.0) and S100A12 (Degree:9.0) were three important up-regulated genes. Previous studies have reported that S100A12 was an important inflammatory marker in coronary heart disease (*Ligthart et al., 2014*; *Wang et al., 2019*), therefore, TLR4 and TREM1 were further studied.

## Basic characteristics of patients

A total of 147 patients were included and completed follow-up. During the half-year follow-up, 48 events happened in AMI patients. There was significant difference in smoking ($p < 0.001$) between two groups. Compared with the normal group, there were higher age, levels of white blood cells (WBC), and fasting plasma glucose (FPG) in the AMI groups ($p < 0.05$), while left ventricular ejected fraction (LVEF) and albumin (ALB) were significantly decreased ($p < 0.005$). The levels of ALT and AST were significantly higher in the AMI group than the normal group ($p < 0.001$). The levels of LDH and CK and DDimer were also measurably higher in the AMI group (Table 2).

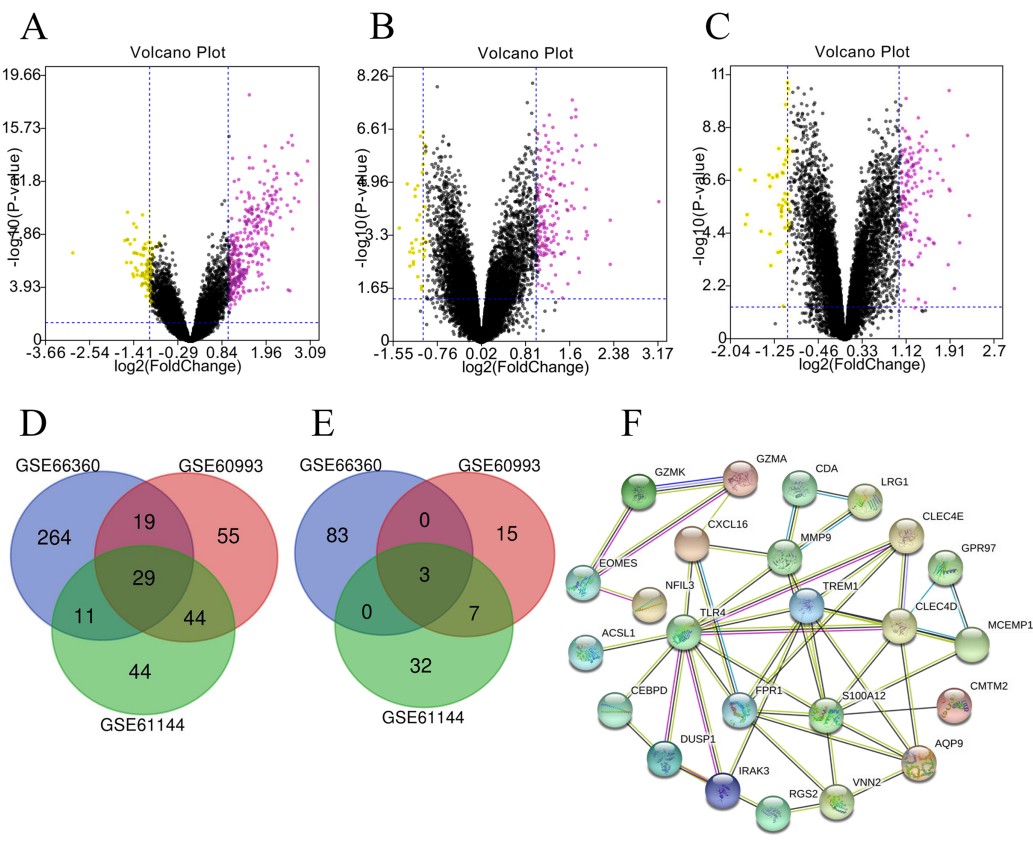

**Figure 2 Candidate genes screening and PPI network construction.** (A/B/C) The volcano plots of GSE66360, GSE60993 and GSE61144 respectively. (D/E) Venn diagram showed 29 upregulated and three downregulated common genes were obtained respectively. (F) PPI network of DEGs.

**Table 1 Common differential expressed genes among GSE66360, GSE61144 and GSE60993.**

| 29 up-regulated DEGs | 3 down-regulated DEGs |
|---|---|
| TLR4 RGS2 CEBPD CLEC4D AQP9 CMTM2 CPD LRG1 GPR97 PYGL MXD1 DUSP1 IRAK3 DYSF CLEC4E IRS2 S100A12 TREM1 RBP7 VNN2 ACSL1 CXCL16 CDA CRISPLD2 NFIL3 CYP4F3 MCEMP1 FPR1 MMP9 | EOMES GZMA GZMK |

## Differential expression of TREM1 and Pearson analysis

According to the bioinformatic results, we examined the expression of TREM1 and TLR4 (Table S1). The result showed no significant difference of TLR4 between two groups. Compared with the normal group, TREM1 expression was markedly increased in the AMI group ($p < 0.001$) (Fig. 3A). Pearson correlation analysis showed that TREM1 expression was positively correlated with age (r = 0.222, $p = 0.007$), neutrophil (r = 0.209, $p = 0.011$), BUN (r = 0.253, $p = 0.003$), UA (r = 0.335, $p < 0.001$), FPG (r = 0.279, $p = 0.001$), HbAC (r = 0.379, $p < 0.001$), BNP (r = 0.411, $p < 0.001$), and left atrial diameter (r = 0.212, $p = 0.014$), and negatively correlated with ALB (r = −0.187, $p = 0.024$) and ApoA1 (r = −0.167, $p = 0.046$) (Table 3). TREM1 expression in the blood was not correlated with monocytes number (r = 0.084, $p = 0.309$) and neutrophil-lymphocyte ratio

**Table 2 Basic characteristics of enrolling patients.**

| Basline characteristics | Normal group ($n = 35$) | AMI group ($n = 112$) | $p$ value |
|---|---|---|---|
| Age (years) | 54.57 ± 10.20 | 61.53 ± 12.26 | 0.030* |
| Gender (male/female) | 12/23 | 83/29 | 0.000** |
| Smoking | 4 | 52 | 0.000** |
| Diabetes | 4 | 38 | 0.010* |
| Hypertension | 14 | 70 | 0.033* |
| BMI (kg/m2) | 25.17 ± 3.70 | 25.37 ± 3.51 | 0.773 |
| Systolic blood pressure (mmHg) | 130.43 ± 14.30 | 131.25 ± 20.52 | 0.826 |
| WBC ($10^9$/L) | 6.16 ± 1.63 | 9.72 ± 3.42 | 0.000** |
| RBC ($10^{12}$/L) | 4.61 ± 0.58 | 4.41 ± 0.64 | 0.108 |
| Hb (g/L) | 138.54 ± 16.67 | 133.90 ± 19.65 | 0.210 |
| ALB (g/L) | 40.74 ± 4.41 | 38.16 ± 3.95 | 0.001** |
| DBil (umol/L) | 3.09 ± 1.23 | 4.12 ± 3.18 | 0.067 |
| ALT (U/L) | 27.34 ± 23.29 | 50.30 ± 43.11 | 0.000** |
| AST (U/L) | 23.23 ± 10.72 | 136.08 ± 151.97 | 0.000** |
| LDH (U/L) | 172.00 (149.25, 200.25) | 289.50 (184.25, 598.75) | – |
| CK (U/L) | 74.50 (55.5, 112.25) | 284.00 (105.00, 1234.00) | – |
| FPG (mmol/L) | 5.98 ± 1.96 | 7.98 ± 3.37 | 0.002* |
| BUN (mmol/L) | 4.98 ± 1.26 | 6.72 ± 4.91 | 0.002* |
| Scr (μmol/L) | 66.00 (55.00, 79.50) | 73.00 (64.00, 86.50) | – |
| UA (μmol/L) | 312.97 ± 95.56 | 351.90 ± 113.14 | 0.093 |
| TG (mmol/L) | 1.62 ± 0.90 | 1.69 ± 1.09 | 0.743 |
| TChol (mmol/L) | 4.60 ± 0.93 | 4.58 ± 1.08 | 0.942 |
| HDL (mmol/L) | 1.24 ± 0.33 | 1.09 ± 0.24 | 0.019* |
| LDL (mmol/L) | 2.78 ± 0.82 | 2.80 ± 0.91 | 0.918 |
| HbAC (%) | 6.02 ± 1.43 | 6.78 ± 1.61 | 0.047* |
| BNP (pg/ml) | 20.00 (7.00, 63.00) | 138.50 (42.30, 530.25) | – |
| TnI (ng/ml) | 0.001 (0.003, 0.01) | 1.435 (0.017, 13.85) | – |
| CRP (mg/L) | 0.82 (0.81, 8.02) | 7.67 (0.82, 20.75) | – |
| LVEF (%) | 0.80 ± 0.60 | 0.56 ± 0.11 | 0.000** |
| INR | 1.06 (1.02, 1.09) | 1.10 (1.05, 1.19) | – |
| Ddimer (μg/L) | 71.00 (47.00, 104.00) | 112.50 (66.25, 232.25) | – |

**Notes:**

* $p < 0.05$.
** $p < 0.01$.

If the data do not satisfy normality and homogeneity of variance (SD ≥ χ), median and quartile (25%, 75%) were calculated, otherwise, the data was described by X ± SD.

RBC, red blood cells; WBC, white blood cells; Hb, hemoglobin; ALT, alanine aminotransferase; AST, aspartate aminotransferase; TC, total cholesterol; TG, total triglyceride; ALB, albumin; DBil, direct bilirubin; BUN, blood urea nitrogen; UA, uric acid; FPG, fasting plasma glucose; HbAC, glycosylated hemoglobin; CRP, C-reactive protein; TG, triglyceride; TChol, total cholesterol; BNP, brain natriuretic peptide and LVEF, left ventricular ejected fraction.

(NLR) (r = 0.128, $p = 0.121$). Variables with great significance in Pearson analysis were included in multiple linear regression. Due to the known correlation between HbAC and FPG, HbAC representing the level of blood glucose control within 2–3 months was selected as one the regression variables. Gensini score ($p = 0.327$) was also selected as a

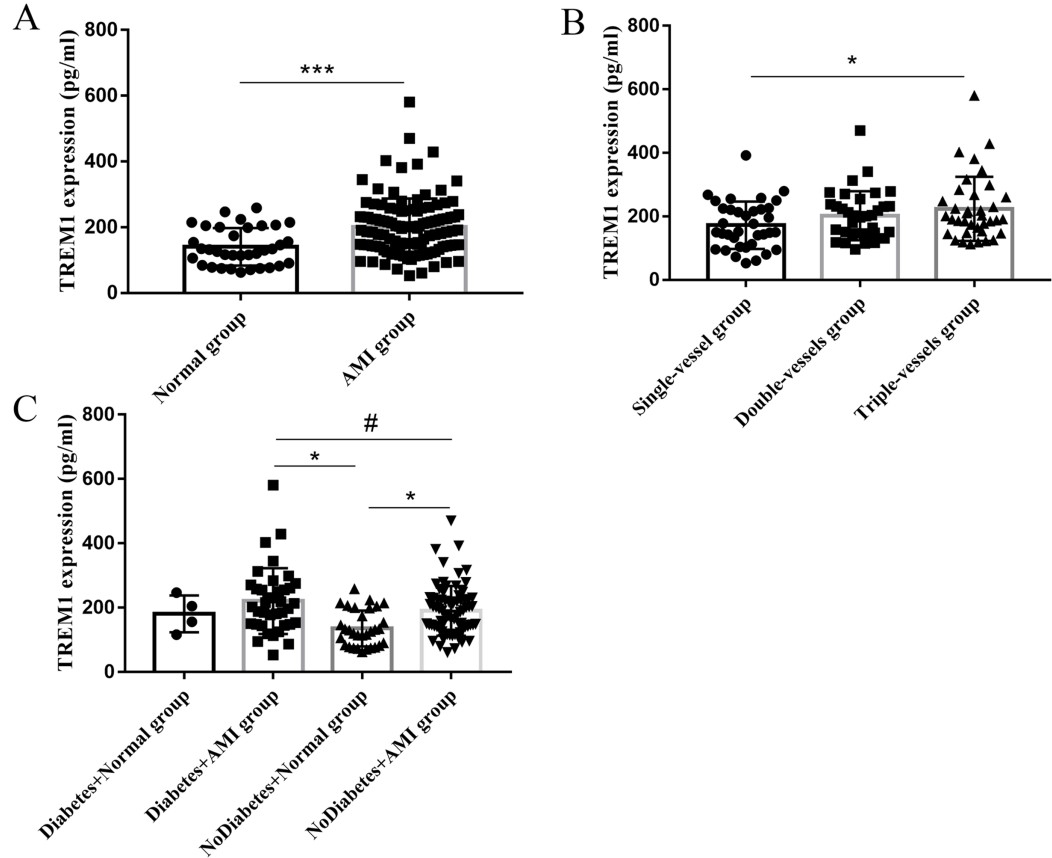

**Figure 3 TREM1 expression in different groups.** (A) TREM1 expression in three groups. Compared with the normal group (140.6 ± 57.0 pg/ml, $n$ = 35), TREM1 expression was markedly increased in AMI group (200.6 ± 87.09 pg/ml, $n$ = 112) ($p$ < 0.001). (B) TREM1 expression in groups with different lesion vessels. TREM1 expression in triple-vessels group (223.9 ± 101.0 pg/ml, $n$ = 37) was significantly higher than that of single-vessel group (172.5 ± 74.5 pg/ml, $n$ = 36) ($p$ < 0.05), the level of which in double-vessels group was 202.7 ± 77.3 (pg/ml, $n$ = 35). (C) TREM1 expression in normal and AMI group with or without diabetes. *$p$ < 0.05, ***$p$ < 0.001, #$p$ = 0.06. 

variable because of its significance representing the severity of coronary lesions. Finally, age, neutrophil, BUN, UA, HbAC, BNP, left atrial diameter, ALB, ApoA1, lesion vessels, and Gensini score were included in the regression. Three methods of linear regression, Stepwise, Forward, and Backward, were all run in SPSS, which showed that BNP and HbAC were two important independent factors influencing TREM1 expression. The fitting equation: $\acute{Y}$ (TREM1) = 0.035 × BNP + 16.520 × HbAC + 66.233 ($R^2$ = 0.283, F = 7.905, $p$ = 0.001). When BNP are addressed, TREM1 was increased by 16.520 units for every one unit increase in HbAC (Table 3).

## TREM1 and subgroup analysis

TREM1 was positively correlated with lesion vessels (r = 0.244, $p$ = 0.011), although it had no correlation with Gensini score ($p$ = 0.327). TREM1 expression in the triple-vessels group (223.9 ± 101.0 pg/ml) was significantly higher than that of the single-vessel group (172.5 ± 74.5 pg/ml) ($p$ < 0.05); the level in the double-vessels group was

**Table 3 Pearson and multilinear regression analysis associated with TREM1.**

| Variables | Pearson analysis | | Multilinear analysis | | |
|---|---|---|---|---|---|
| | r | p | β | t | p |
| Age (years) | 0.222 | 0.007** | | | |
| Neutrophil ($10^9$/L) | 0.209 | 0.011* | | | |
| BUN (mmol/L) | 0.253 | 0.003** | | | |
| UA (μmol/L) | 0.335 | <0.001** | | | |
| HbAC (%) | 0.379 | <0.001** | 16.520 | 1.979 | 0.055 |
| BNP (pg/ml) | 0.411 | <0.001** | 0.035 | 2.922 | 0.006** |
| LA (mm) | 0.212 | 0.014* | | | |
| ALB (g/L) | −0.187 | 0.024* | | | |
| ApoA1 (g/L) | −0.167 | 0.046* | | | |
| Vessels | 0.244 | 0.011* | | | |
| Gensini score | 0.095 | 0.327 | | | |

Notes:
* $p < 0.05$.
** $p < 0.01$.
Significant variables with no internal relation in pearson analysis were entered into multilinear analysis, $p < 0.1$ was set for entering variables to avoid losing significant factors. Gensini score representing the severity of CAD was also included in analysis. Methods of "Stepwise", "Forward" and "Backward" in multiple linear regression were all performed by SPSS, which showed that BNP and HbAC were two important independent factors influencing TREM1 expression. Finally, the fitting equation: $\hat{Y}$ (TREM1) = 0.035 × BNP + 16.520 × HbAC + 66.233 ($R^2$ = 0.283, F = 7.905, $p$ = 0.001).
BUN, blood urea nitrogen; UA, uric acid; HbAC, glycosylated hemoglobin; BNP, brain natriuretic peptide; LA, left atrium and ALB, albumin.

202.7 ± 77.3 (pg/ml) (Fig. 3B). TREM1 expression showed a significant difference in the normal and AMI groups in diabetes subgroups (Fig. 3C). Compared to the normal group without diabetes, TREM1 expression was higher in the AMI group with or without diabetes ($p < 0.05$). In the AMI group, TREM1 expression was higher in patients with diabetes than the group without diabetes, although it showed no significant difference ($p = 0.06$).

## TREM1 and diagnosis of AMI

The ROC curve showed that TREM1 played a novel role in the diagnosis of AMI ($p < 0.001$) when the area under the curve (AUC) was 0.721, with a sensitivity of 0.786 and a specificity of 0.600 (Yoden Index = 0.386) (Fig. 4A). In AMI groups without diabetes, the diagnostic sensitivity was 0.757, the specificity was 0.631 and the AUC was 0.724 (Yoden Index = 0.402, $p < 0.001$) (Fig. 4B). It seemed that TREM1 showed more significant diagnostic value in AMI patients without diabetes mellitus.

## TREM1 and prognosis of AMI

The role of TREM1 in the 6-month follow-up was also studied. MACEs such as rehospitalization because of recurrent angina pectoris and heart failure, revascularization, recurrent MI, cardiac death, and total death were all recorded. Patients with AMI were divided into high risk and low risk groups according to the median of TREM1 (high risk ≥ 177.5, low risk < 177.5). In the Kaplan-Meier analysis, the incidence of MACEs in the high risk group was measurably higher than in the low risk group ($p < 0.0001$). Then, Cox

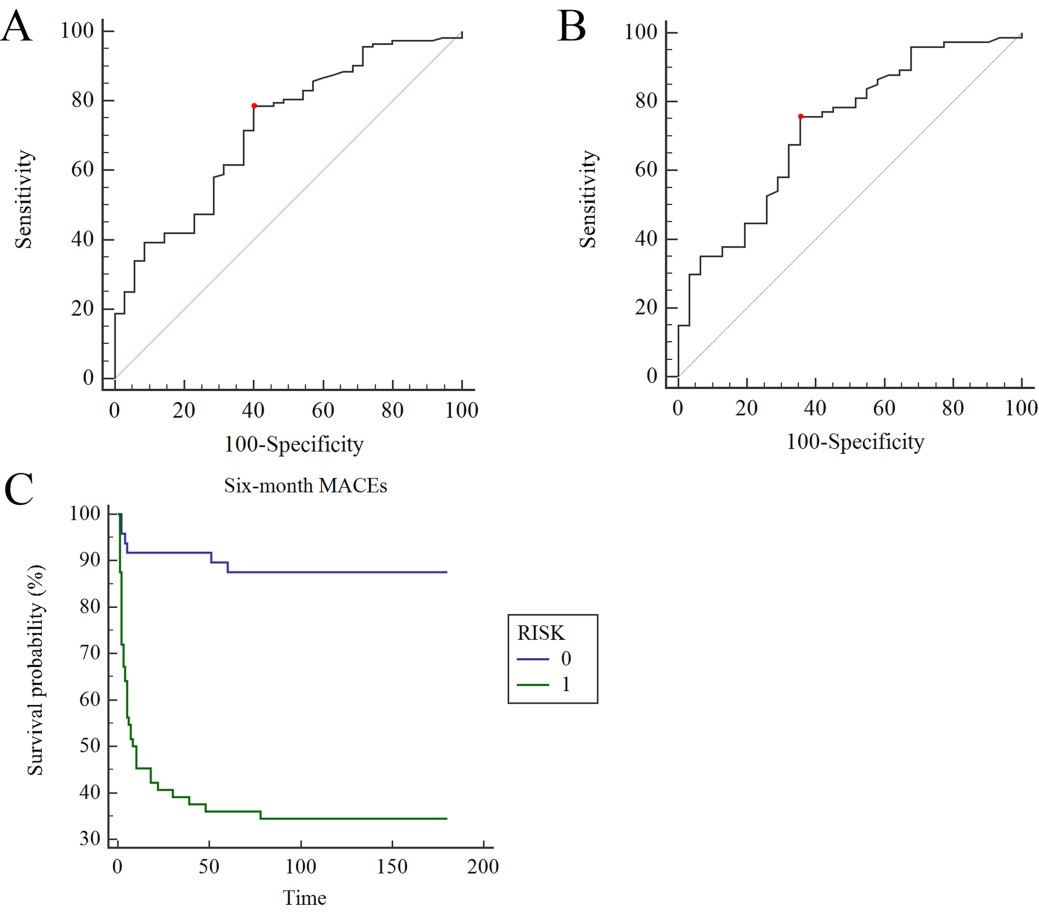

**Figure 4 The diagnosis and prognosis value of TREM1 in AMI.** (A) In AMI group ($p < 0.001$), the AUC was 0.721, with a sensitivity of 0.786 and a specificity of 0.600 (Yoden Index = 0.386). (B) In AMI groups without diabetes, the diagnostic sensitivity was 0.757, the specificity was 0.631 and the AUC was 0.724 (Yoden Index = 0.402, $p < 0.001$). (C) In Kaplan–Meier analysis, incidence of MACEs in high risk group was obviously higher than that in low risk group ($p < 0.0001$). 0, low risk group; 1, high risk group.

regression analysis was performed to explore the important variables affecting MACE incidences (Table 4). TREM1 ($p < 0.001$), heartrate ($p = 0.001$), RBC ($p = 0.048$), Hb ($p = 0.019$), CK ($p = 0.022$), FPG ($p = 0.021$), BUN ($p = 0.013$), UA ($p = 0.044$), LA ($p = 0.05$), IVS ($p = 0.031$), LV ($p = 0.020$), LVEF ($p = 0.001$), and vessels ($p = 0.040$) screened by univariate Cox regression analysis were entered into the variate Cox analysis. Finally, we determined that TREM1, Hb and IVS were three important independent factors affecting the 6-month outcome of AMI patients (F = 34.091, $p < 0.001$).

## DISCUSSION

We used bioinformatics to search gene expression profiles of AMI to explore the DEGs of AMI. According to Cytoscape, TREM1 and TLR4 were considered two of the most important genes of AMI, but only TREM1 showed a significant difference. We found that TREM1 may be helpful for the diagnosis of MI and is associated with the number of lesion vessels. The more novel findings were that TREM1 was closely associated with diabetes

**Table 4 Univariate and variate Cox regression analysis for predicting 6-month MACEs after AMI.**

| Variables | Unvariate analysis | | | Variate analysis | | |
|---|---|---|---|---|---|---|
| | Exp (β) | Wald | p | Exp (β) | Wald | p |
| TREM1 (pg/ml) | 1.007 | 35.535 | <0.001** | 1.006 | 23.411 | 0.000** |
| Heartrate | 1.029 | 10.479 | 0.001** | | | |
| RBC ($10^{12}$/L) | 0.656 | 3.896 | 0.048* | | | |
| Hb (g/L) | 0.984 | 5.543 | 0.019* | 0.981 | 7.459 | 0.006** |
| CK (U/L) | 1.000 | 5.246 | 0.022* | | | |
| FPG (mmol/L) | 1.092 | 5.334 | 0.021* | | | |
| BUN (mmol/L) | 1.052 | 6.216 | 0.013* | | | |
| UA (μmol/L) | 1.002 | 4.049 | 0.044* | | | |
| LA (mm) | 1.874 | 3.826 | 0.050 | | | |
| IVS (mm) | 1.613 | 4.680 | 0.031* | 5.505 | 4.704 | 0.030* |
| LV (mm) | 1.824 | 5.453 | 0.020* | | | |
| LVEF (%) | 0.013 | 10.849 | 0.001** | | | |
| Vessels | 1.461 | 4.229 | 0.040* | | | |

Notes:
* $p < 0.05$.
** $p < 0.01$.
Significant variables in univariate analysis were entered into multiple Cox regression and only TREM1 was screened out as a significant variable, which indicated that TREM1 was an independent factor predicting 6-month MACEs.
RBC, red blood cells; Hb, hemoglobin; CK, creatine kinase; BUN, blood urea nitrogen; UA, uric acid; FPG, fasting plasma glucose; LA, left atrium and LVEF, left ventricular ejected fraction.

mellitus and it was an independent predictor for 6-month MACEs after AMI. There was higher incidence of MACEs in patients with a higher TREM1 level.

As a transmembrane glycoprotein, TREM1 can bind with a transmembrane receptor DAP12 (DNX-activating protein 12) to form a TREM1/DAP12 complex (*Lanier, 2009*; *Ormsby et al., 2011*; *Tessarz & Cerwenka, 2008*). Immunoreceptor tyrosine activation motif (ITAM) tyrosine of DAP12 is phosphorylated by the protein tyrosine kinase of the SRC family, so that the CBL and growth factor receptor binding protein 2 (Grb 2) is phosphorylated. Phospholipid polymyo-inositol 3 kinase (PI3K) and extracellular signal-regulated kinase (ERK) pathways were further activated, to regulate calcium homeostasis, induce activation of transcription factors, and promote the production of pro-inflammatory cytokines and adhesion molecules (*Colonna, 2003*; *Klesney-Tait et al., 2013*; *Lanier, 2009*; *Colonna & Facchetti, 2003*; *Pelham & Agrawal, 2014*; *Turnbull & Colonna, 2007*). TREM1 mainly expressed in myeloid lineage cells, especially in monocytes/macrophages, and it was involved in the innate immune response and amplified proinflammatory response in both infectious and non-infectious diseases (*Liu et al., 2019*; *Pelham & Agrawal, 2014*). TREM1 was closely related to Crohn disease and inflammatory bowel disease (IBD) (*Chapuy et al., 2019*; *Verstockt et al., 2019*) and sepsis (*Gibot et al., 2004*). For sterile inflammatory diseases, it was reported that TREM1 played a critical role in the pathogenesis of Alzheimer's disease (*Saadipour, 2017*) because of the potential function in innate immune response and DNA methylation (*Sao et al., 2018*). TREM1 could also predict the prognosis of renal cell carcinoma (*Yamada et al.,*

*2018*). TREM1-knockout markedly decreased inflammation and oxidative stress in mice with spinal cord injury, along with the decreased expression of TLR2 and TLR4 (*Li et al., 2019*). Therefore, potential mechanisms of TREM1 in AMI may be the synthesis of inflammation amplification and the following imbalance of oxidative stress. Although most studies about TREM1 all focused on inflammatory diseases such as IBD, we cannot ignore its role in cardiovascular disease due to emerging immune mechanisms in MI (*Pelham & Agrawal, 2014*).

Univariate analysis showed that TREM1 was positively correlated with BUN, UA, BNP, and LA; and negatively correlated with ALB and ApoA1. As a diagnostic biomarker of heart failure, BNP can be combined with BUN and UA to predict the prognosis of heart failure (*Oki et al., 2019*; *Testani et al., 2014*). Increased LA was also associated with the occurrence of AF and acute heart failure (*Oikonomou et al., 2019*; *Van Aelst et al., 2018*), while low ALB level was an important predictor for poor prognosis after cardiac surgery (*Van Beek et al., 2018*). ApoA1 was an apolipoprotein of HDL which has anti-inflammatory and antioxidant effects (*Gombos et al., 2017*). TREM1 was related to these markers associated with adverse outcome of cardiac diseases, so we hypothesized that TREM1 may also be associated with the prognosis of myocardial infarction, which was also confirmed by the follow-up results.

Interestingly, our study showed TREM1 was positively associated with HbAC and FPG; and TREM1 expression was significantly increased in patients with both AMI and diabetes compared to those with only AMI. A study pointed out that sTREM1 expression was significantly higher in patients with obesity and diabetes than those with only obesity or without obesity and diabetes, which points to the important role of TREM1 in the potential pathophysiology of obesity and diabetes (*Subramanian et al., 2017a*). TREM1 overexpression was also related to insulin resistance induced by obesity (*Subramanian et al., 2017b*). A case-control study from Denmark showed sTREM1 level was elevated in children with newly diagnosed type 1 diabetes compared to their siblings (*Thorsen et al., 2017*). Thus, TREM1 may play an important role in both acute myocardial infarction and diabetes progression.

We also found that TREM1 was positively related to the number of lesion vessels, although there was no relationship between TREM1 and Gensini score. TREM1 expression in the triple-vessels group was notably higher than other groups. Apart from this, TREM1 could predict 6-month MACEs after AMI. It has been reported that TREM1 was involved in the pathogenesis of atherosclerosis. Jeremie Joffre et al used TREM1-knockdown mice and TREM1 inhibitor (LR12 peptide) to confirm that TREM1 can promote plaque inflammation and the formation of foam cells, furthermore, TREM1 expression was higher in lipid-rich plaques than that of fibrous plaques (*Huynh, 2017*; *Joffre et al., 2016*). High-fat induced high expression of TREM1 in myeloid cells promoted generation of inflammatory factors and formation of foam cells (*Zysset et al., 2016*). TREM1 participated in the mechanism of pravastatin improving atherosclerosis (*Wang, Gao & Lu, 2018*). Therefore, increased TREM1 expression may become a potential predictor of vulnerable plaques.

There are still some limitations in this study. First, the follow-up period is only 6 months and incidence of MACE is low. Second, this study only detected the expression level of TREM1 in the plasma of patients with AMI on admission, but did not dynamically monitor the changes of TREM1 one week after myocardial infarction. A follow-up study can focus on the dynamic expression profile of TREM1 after myocardial infarction, to provide a theoretical basis for TREM1 to participate in myocardial infarction risk stratification.

In conclusion, using bioinformatics and after verification using plasma samples of patients with AMI, we found TREM1 is an important biomarker in myocardial infarction. TREM1 was also related to the number of lesion vessels and closely associated with diabetes. TREM1 could also predict the 6-month MACEs after AMI. Therefore, TREM1 would be a novel potential predictor for adverse events after MI.

## ACKNOWLEDGEMENTS

We thank Professor King Lu and Doctor Wang Shiyuan from School of Public Health, Southeast University for statistical support.

### Funding

This work was funded by the Jiangsu Provincial Key Research and Development Program BE2016785 and the Jiangsu Provincial Key Medical Discipline (Laboratory ZDXKA2016023). The funders had no role in study design, data collection and analysis, decision to publish, or preparation of the manuscript.

### Grant Disclosures

The following grant information was disclosed by the authors:
Jiangsu Provincial Key Research and Development Program: BE2016785.
Jiangsu Provincial Key Medical Discipline: ZDXKA2016023.

### Competing Interests

The authors declare that they have no competing interests.

### Author Contributions

- Zhenjun Ji conceived and designed the experiments, performed the experiments, analyzed the data, prepared figures and/or tables, authored or reviewed drafts of the paper, and approved the final draft.
- Rui Zhang conceived and designed the experiments, performed the experiments, analyzed the data, prepared figures and/or tables, authored or reviewed drafts of the paper, and approved the final draft.
- Mingming Yang conceived and designed the experiments, performed the experiments, prepared figures and/or tables, authored or reviewed drafts of the paper, and approved the final draft.

- Wenjie Zuo performed the experiments, authored or reviewed drafts of the paper, and approved the final draft.
- Yuyu Yao conceived and designed the experiments, performed the experiments, authored or reviewed drafts of the paper, and approved the final draft.
- Yangyang Qu performed the experiments, authored or reviewed drafts of the paper, and approved the final draft.
- Yamin Su performed the experiments, authored or reviewed drafts of the paper, and approved the final draft.
- Zhuyuan Liu performed the experiments, authored or reviewed drafts of the paper, and approved the final draft.
- Ziran Gu performed the experiments, analyzed the data, prepared figures and/or tables, authored or reviewed drafts of the paper, and approved the final draft.
- Genshan Ma conceived and designed the experiments, authored or reviewed drafts of the paper, and approved the final draft.

## Human Ethics

The following information was supplied relating to ethical approvals (i.e., approving body and any reference numbers):

This study was approved by the Ethics Committee of Zhongda Hospital Southeast University (2018ZDSYLL134-P01).

## Data Availability

The raw data are available as a Supplemental File.

## Supplemental Information

Supplemental information for this article can be found online at http://dx.doi.org/10.7717/peerj.11655#supplemental-information.

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
