# Peer review of "Accuracy of triggering receptor expressed on myeloid cells 1 in diagnosis and prognosis of acute myocardial infarction: a prospective cohort study"

_PeerJ, doi:10.7717/peerj.11655_

## Round 0.1 · original submission · Major Revisions

I recommend you carefully consider all criticisms and comments raised by the Reviewers.

Reviewer 1 ·

Basic reporting

1. I recommend changing the title of the study be like:
"Accuracy of Triggering Receptor Expressed on Myeloid Cells 1 in Diagnosis and Prognosis of Acute Myocardial Infarction: Retrospective Cohort Study"
2. The paper requires language editing.
3. Introduction:
 Line number 45, using the "MACEs" abbreviation is not acceptable, the abbreviation should be defined previously.
 Line number 46, using the "MI" abbreviation is not acceptable, the abbreviation should be defined previously.
 The gab of knowledge is not clearly outlined in the introduction; the authors should explain the gab clearly.

Experimental design

4. I suggest the method section be structured based on Strengthening the Reporting of Observational Studies in Epidemiology (STROBE) Statement: https://www.strobe-statement.org/index.php?id=available-checklists
5. From line number 77 to 81, the distribution of patient and healthy persons in groups "GSE66360, GSE61144, and GSE60993" is not clearly identified.
6. Actually which statistical software is used SPSS or R studio? I'm confused that authors say in the abstract they used R software and in the Statistical analysis part of the Method Authors say they used SPSS.
7. Authors should regulate any online tools they used in the method section clearly.

Validity of the findings

8. Table 2, 3, and 4: the authors should define all the abbreviations mentioned in any of the tables below them.
9. Table 1, 2, 3, and 4: the authors should define all the units of measurement.
10. Providing more interpretation of results in the discussion section with more evidence in comparing your results with the previous studies.
11. Providing of the limitations, strength, and recommendations sections.
12. I suggest the result section be structured based on Strengthening the Reporting of Observational Studies in Epidemiology (STROBE) Statement: https://www.strobe-statement.org/index.php?id=available-checklists

Additional comments

The paper requires language editing.
Small sample size.

Reviewer 2 ·

Basic reporting

Authors of the paper named “Diagnosis and prognosis value of Triggering Receptor Expressed on Myeloid Cells 1 in patients with acute myocardial infarction” aimed to examine whether this protein can be used as a novel biomarker of AMI. Using the bioinformatics, authors have examine differentially expressed genes between healthy control individuals and AMI patients. TREM-1 was singled out as a significantly upregulated gene in AMI patients. To confirm results obtained analysing GEO, authors have measured TREM-1 levels in plasma samples of 147 individuals (35 healthy controls and 112 AMI patients), obtained 4h after admission. Plasma levels of TREM-1 were significantly upregulated in AMI patients. Upregulation of TREM-1 was also corelated with the severity of CAD as the triple vessel group had noticeably higher levels than single vessel patients. At the same time hyperglycaemia seemed not to have a significant impact on TREM-1 expression. Finally, Cox regression analysis of the patients involved singled out TREM-1 next to Hb and IVS as a factor that strongly affects 6-month outcome in AMI patients.

Experimental design

General comment:
Scientific question is well formulated. It is methodically and logically well formulated paper, easy to follow and with a clear message. The experimental design is suitable to address the raised question.

Validity of the findings

Impression of this reviewer is that some of the conclusions are not fully supported with the data

Additional comments

Major comment:
Authors have concluded that TREM-1 can be indeed used as biomarker. As it is written in the text of the manuscript, TREM-1 is expressed on myeloid cells (neutrophils, monocytes, macrophages). These cells are the main responders to the cardiac ischaemic injury. Number of invading neutrophils and monocytes during the inflammatory phase directly corelates with the tissue damage. From the presented data, it seems that that TREM-1 is actually a marker of neutrophil/monocyte recruitment. Any state that elevates number of these cells will also elevate levels of plasma TREM-1. Therefore, can we really say that TREM-1 is an AMI biomarker??? It would be helpful if the authors would show TREM-1 levels in the later time points.
Additionally, it is not clear why authors have chose TREM-1 at the same time ignoring the S100A12 when this gene was also strongly upregulated in AMI patients?
This reviewer finds a bit strange that other S100 proteins didn’t come up in bioinformatics data, namely S100A8 and S100A9. Can you comment on this?
Minor comment:
1. Is TREM-1 corelated with neutrophils and monocytes numbers in the blood?
2. Can you please provide NLR (neutrophil-lymphocyte ratio) upon admission and how this number corelates with TREM-1 plasma levels?
3. Introduction, line 66 “still remains further” please change into “still requires further”
4. Introduction, line 69, please rephrase the sentence.

---

## Round 0.2 · accepted · Accept

The Reviewer has evaluated your revised manuscript favorably. Hence, it can be accepted for publication in PeerJ.

Reviewer 1 ·

Basic reporting

'no comment'

Experimental design

'no comment'

Validity of the findings

'no comment'

Additional comments

The article is acceptable for me.
Thanks for the respectful following of the reviewers' comments.